# GDNF Increases Inhibitory Synaptic Drive on Principal Neurons in the Hippocampus via Activation of the Ret Pathway

**DOI:** 10.3390/ijms232113190

**Published:** 2022-10-29

**Authors:** Apostolos Mikroulis, Eliška Waloschková, Johan Bengzon, David Woldbye, Lars H. Pinborg, Bo Jespersen, Anna Sanchez Avila, Zsofia I. Laszlo, Christopher Henstridge, Marco Ledri, Merab Kokaia

**Affiliations:** 1Experimental Epilepsy Group, Department of Clinical Sciences, Faculty of Medicine, Lund University, Sölvegatan 17, BMC A11, 22362 Lund, Sweden; 2Epilepsy Center, Department of Clinical Sciences, Faculty of Medicine, Lund University, Sölvegatan 17, BMC A11, 22362 Lund, Sweden; 3Neurosurgery, Department of Clinical Sciences, Faculty of Medicine, Lund University Hospital, Skånes Universitetssjukhus, 22185 Lund, Sweden; 4Lund Stem Cell Center, Department of Clinical Sciences, Faculty of Medicine, Lund University, Klinikgatan, 22362 Lund, Sweden; 5Department of Neuroscience, University of Copenhagen, Blegdamsvej 3B, DK-2200 Copenhagen, Denmark; 6Epilepsy Clinic and Neurobiology Research Unit, Department of Neurology, Copenhagen University Hospital, Rigshospitalet, Copenhagen, Inge Lehmanns Vej 6-8, Rigshospitalet, DK-2100 Copenhagen, Denmark; 7Department of Neurosurgery, Copenhagen University Hospital, Rigshospitalet, Inge Lehmanns Vej 6, Rigshospitalet, DK-2100 Copenhagen, Denmark; 8Jacqui Wood Centre, Division of System Medicine, School of Medicine, University of Dundee, Dundee DD1 9SY, Scotland, UK; 9Laboratory of Molecular Neurophysiology and Epilepsy, Department of Clinical Sciences, Faculty of Medicine, Lund University, Sölvegatan 17, BMC A11, 22362 Lund, Sweden

**Keywords:** GDNF, epilepsy, ret, IPSC, electrophysiology

## Abstract

Glial cell line-derived neurotrophic factor (GDNF) has been shown to counteract seizures when overexpressed or delivered into the brain in various animal models of epileptogenesis or chronic epilepsy. The mechanisms underlying this effect have not been investigated. We here demonstrate for the first time that GDNF enhances GABAergic inhibitory drive onto mouse pyramidal neurons by modulating postsynaptic GABA_A_ receptors, particularly in perisomatic inhibitory synapses, by GFRα1 mediated activation of the Ret receptor pathway. Other GDNF receptors, such as NCAM or Syndecan3, are not contributing to this effect. We observed similar alterations by GDNF in human hippocampal slices resected from epilepsy patients. These data indicate that GDNF may exert its seizure-suppressant action by enhancing GABAergic inhibitory transmission in the hippocampal network, thus counteracting the increased excitability of the epileptic brain. This new knowledge can contribute to the development of novel, more precise treatment strategies based on a GDNF gene therapy approach.

## 1. Introduction

### 1.1. GDNF

The glial cell line-derived neurotrophic factor (GDNF) was initially identified as a survival factor for dopaminergic neurons [1]. Other members of the GDNF family include neurturin, artemin, and persephin, all of which bind to their selective GFRα receptors (GFRα1, 2, 3, and 4). The GDNF and GFRα1 together activate the transmembrane receptor tyrosine kinase (Ret) to induce intracellular signaling [2]. GDNF and GFRα1 can also signal in a Ret-independent way through neural cell adhesion molecule NCAM [3]. In addition, GDNF, in combination with GFRα1, can act independently of Ret and NCAM and function as a cell adhesion molecule, a process termed ligand-mediated cell adhesion (LICAM) [4], contributing to synapse formation in the hippocampus. Another receptor for GDNF has been identified as the heparin sulfate proteoglycan Syndecan-3 [5].

GDNF is widely distributed in the rodent and human central nervous systems [6,7]. In the hippocampus, an important area for seizure generation, high expression of GDNF is found in hippocampal pyramidal cells and dentate gyrus (DG) granule cells [8]. Moreover, the GDNF-specific GFRα1 and NCAM are co-expressed in the same cells within the rat hippocampus [4,9].

### 1.2. Epilepsy

Epilepsy is a devastating neurological condition affecting over 60 million people worldwide. Current anti-seizure medications (ASMs) provide only symptomatic relief, have multiple associated side effects, and are ineffective in 30–40% of the cases [10,11]. Developing novel treatment strategies for particularly drug-resistant epilepsy is thus urgently needed.

The hallmark of epilepsies is abnormal, highly synchronized activity of neurons that often starts in a limited brain area and then may spread to other parts of the brain. This is thought to be caused by increased excitability of the local neuronal circuits caused by excessive excitatory synaptic activity or/and decreased inhibitory synaptic activity [12]. The initial precipitating event leading to such hyper-excitability could be traumatic brain injury (TBI), stroke, blood–brain barrier (BBB) disruption, inflammation, brain tumor, genetic predisposition, etc., but often the reason is not known [13]. As studies on animals suggest, neuronal network excitability, which undergoes dynamic changes over time, has to increase to a certain threshold level to generate a seizure. Such an assumption would imply that in people with epilepsy increased network excitability can reach the threshold for seizure generation more easily. If this is true, decreasing network excitability permanently by any means would prevent seizures and thus cure the disease. Indeed, many ASMs that are used today decrease brain excitability and thereby prevent seizure generation. However, these drugs have only short-lasting effects and need to be taken regularly, often causing systemic side effects and failing to control seizures in one-third of patients.

### 1.3. GDNF and Epilepsy

Based on our previous studies in naive animals, a gain-of-function approach with viral vector-based overexpression of GDNF in the hippocampus exerts an inhibitory effect on acute seizures in electrical kindling [14] or status epilepticus (SE) models [15]. In these studies, bilateral intrahippocampal rAAV-based GDNF gene delivery or bilateral encapsulated cell biodelivery were used, respectively. Moreover, the latter approach was more recently tested in chronic animal models of epilepsy [16,17], demonstrating an inhibitory effect on spontaneous recurrent seizures. Thus, GDNF becomes an interesting candidate to investigate as a novel treatment alternative for chronic focal epilepsies in humans. However, before considering this avenue, the mechanisms by which GDNF inhibits seizures need to be clarified. Although several hypotheses have been put forward, the current understanding of seizure-suppressant mechanisms of GDNF is rather limited. One possibility is that GDNF promotes the survival of inhibitory interneurons, similar to what has been shown for dopaminergic neurons in the substantia nigra [18] or in molecular layer interneurons of the cerebellum [19]. Alternatively, GDNF might promote inhibition indirectly by other mechanisms, such as stimulating neurite outgrowth [20] or inhibiting microglia activation [21]. GDNF may exert an effect through Ret [2], NCAM [3], or Syndecan-3 [5] pathways, but which of these are involved in its seizure-suppressant effect is currently unknown.

Here, by using a combination of electrophysiology, Western blot, and imaging techniques, we first demonstrate that elevated extracellular levels of GDNF increase inhibitory synaptic drive on principal neurons in mouse and human acute hippocampal slices. We then provide evidence that these effects are both pre- and postsynaptic and that GDNF acts preferentially via a Ret-dependent pathway.

## 2. Results

### 2.1. GDNF Enhances Inhibitory Inputs to CA1 Pyramidal Neurons

To investigate whether GDNF can directly alter synaptic transmission in the mouse hippocampus, we performed electrophysiological recordings and measured postsynaptic currents from CA1 pyramidal neurons. Whole-cell recordings from hippocampal slices incubated with 2 nM GDNF demonstrated an increased frequency of spontaneous and miniature inhibitory postsynaptic currents (sIPSCs and mIPSCs, representative traces are shown in Figure 1A,B, properties in Table 1 and Appendix A) as compared to controls (Figure 1C), and a corresponding decrease in inter-event intervals (IEI) as demonstrated by cumulative probability curves (Figure 1C). The amplitude of the sIPSCs was also increased, as shown by the analysis of median values based on individual cells. The cumulative probability curves demonstrated complex changes, with curves crossing each other twice (Figure 1D). The sIPSCs with lower amplitudes (less than 20 pA) and those with higher amplitudes (over 30 pA) increased in magnitude, while intermediate ones (between 20 and 30 pA were slightly decreased. The increase in mIPSC amplitudes in the GDNF-treated group, as analyzed by cell-based averages, did not reach statistical significance (Figure 1D) but was statistically significant according to cumulative probability curves (*p* < 0.01 and D > 0.05), which, in contrast to sIPSCs, demonstrated that amplitudes below 30 pA were increased, while those over 30 pA were decreased in the magnitude. Taken together, these data suggest that both the frequency and amplitude of IPSCs were increased in slices exposed to GDNF. These changes were not present for excitatory postsynaptic currents (Appendix A).

Interestingly, we also observed a marked decrease in the rise times of both sIPSCs and mIPSCs with cell-based average analysis and cumulative probability curves (Table 1; Figure 1B,E). The latter results prompted further analysis of the relationship between IPSC amplitudes and the corresponding rise times of the events. We found an inverse amplitude-rise time relation for the entire sample of events, with higher amplitude events also having a faster rise time than lower amplitude events. This observation indicates passive filtering of synaptic events since currents that originate further away from the recording site (in the dendritic tree) will be passively low-pass filtered as well as having a lower apparent amplitude due to axial resistivity and ionic diffusion [22].

To examine whether these IPSCS with various kinetics were differentially affected by GDNF exposure, we classified the events according to their rise-time values in “fast” and “slow” groups, with a cut-off around a 1.3 ms threshold (see methods). We then counted the number of events in each group and compared the proportion of fast and slow rise time events between the control and GDNF-incubated groups (Figure 2, scatter- and pie charts). The GDNF-incubated group had proportionally more fast than slow events compared to the controls (Fisher’s exact *p* < 0.001, both for sIPSCs and mIPSCs).

Taken together, these data suggest that GDNF induced changes both in frequency and amplitude of IPSCs. In addition, GDNF increased the proportion of high amplitude/fast rise time events, suggesting increased perisomatic inhibitory drive onto pyramidal neurons.

### 2.2. Inhibitory Synapse Density Is Increased in the Pyramidal Layer

To further investigate the pre- vs. postsynaptic site of GNDF action and the specific contribution of perisomatic inhibitory synapses from parvalbumin (PV) interneurons onto CA1 pyramidal cells, we performed additional electrophysiological investigations taking advantage of optogenetic tools for specific activation of PV interneurons with Channelrhodopsin-2 (ChR2). We incubated slices obtained from a PV-ChR2 transgenic mouse line with GDNF or control solution for 1 h and recorded from CA1 pyramidal cells as previously. During the recording, we stimulated GABA release from PV inhibitory terminals by exposing the slices to 20 light pulses at a frequency of 20 Hz and measured light-evoked postsynaptic currents (lePSCs). By averaging responses from 10 consecutive trains obtained at 15 s intervals, we calculated releasable pool and release probability estimates at PV synapses (see methods), and observed no difference between GDNF and control incubated slices (for normalised pool estimates: Ctrl 1.73 ± 0.18, n = 8; GDNF 1.49 ± 0.10, n = 8; for release probability estimates: Ctrl 0.47 ± 0.03, n = 7; GDNF 0.48 ± 0.02, n = 7, Appendix A). These data suggest that the pre-synaptic release of GABA, at least from PV interneurons, is not affected by GDNF incubation.

As an additional investigation on the GDNF site of action, we performed double immunostainings for gephyrin, a scaffolding protein for glycine and GABA_A_ receptors [23], and PV or GAD65/67. The part of the gephyrin-stained puncta associated with the cell nucleus staining (Hoescht staining, blue, Figure 3A) reflected intracellular gephyrin [24] and not synaptic localization and was therefore excluded from the analysis. Quantification of the number of remaining gephyrin staining puncta, using the same exposure time and laser settings during image acquisition at the confocal microscope, revealed an increase in gephyrin staining density in the pyramidal layer of GDNF-incubated slices, compared to controls (Figure 3B). However, no significant differences in either PV or GAD65/67 staining densities were detected between control and GDNF-incubated slices (Figure 3B), indicating that the number of GABAergic pre-synapses was not changed by GDNF incubation. This finding supports the postsynaptic site of GDNF action.

### 2.3. GDNF Effect Is Mediated by Ret Pathway Activation

Next, we investigated potential intracellular pathways responsible for the observed effect of GDNF. GDNF binds with high affinity to the soluble GFRα1 co-receptor and subsequently to Ret [2]. The involvement of the Ret pathway was tested by applying XIB4035, a positive modulator of the GFRα1/Ret pathway [25], to the hippocampal slices. Preincubation with XIB4035 (10 µM), in combination with GDNF, decreased IEI and increased amplitudes of both sIPSCs and mIPSCs compared to GDNF alone when analyzed by cumulative probability curves (Figure 4A; *p* < 0.01 and D > 0.05, Table 2 and Appendix A). A similar trend was also seen in the cell-based analysis of averaged values for frequency and amplitude.

To further confirm the involvement of the Ret pathway, we tested whether GDNF incubation increases the levels of activated (phosphorylated) Ret, using Western blots on extracted protein from slices treated identically to the electrophysiology experiments. Comparing the ratio of phosphorylated Ret to total Ret (Figure 5) demonstrated a significant relative increase in phosphorylated Ret in slices treated with GDNF (1.238 ± 0.028, n = 4) as compared to controls. However, the Ret phosphorylation was not further increased in XIB4035 + GDNF treated samples (1.169 ± 0.032, n = 4), suggesting that the phosphorylation reached its maximum by GDNF treatment alone. Addition of the Ret inhibitor SPP86 together with GDNF reverted Ret phosphorylation to control levels (1.014 ± 0.047, n = 4). Overall, these results suggest that the GDNF effect is mediated by Ret pathway activation and its downstream signaling.

### 2.4. Potential Involvement of Syndecan3 Pathway in GDNF Effect

Since GDNF induces changes in inhibitory neurotransmission, it is plausible that its effect can also be mediated by Syndecan3, which has been reported to be expressed in inhibitory interneurons [5]. However, the postsynaptic changes in immunostainings showing increased GABA_A_ receptor clustering would only be possible if Syndecan3 is also expressed in excitatory principal neurons of CA1 [26]. To confirm the Syndecan3 expression site, we performed a detailed analysis with an array tomography approach (Figure 6). Based on the pattern and orientation of the immunostainings, Syndecan3 was primarily expressed in prolonged axonal segments in the stratum radiatum of the CA1 area of the hippocampus (Figure 6E–H). This staining pattern has been described previously for Syndecan3, suggesting the protein is located on the incoming axons of the Schaffer collaterals [27]. Alternatively, these segments are axons of inhibitory interneurons. These immunostained segments were not co-localized with synaptophysin immunostaining, suggesting the expression of Syndecan3 in axons but not synaptic terminals. In fact, the co-localization analysis showed that only 0.07 ± 0.04% of Synaptophysin objects co-localized with Syndecan3, and conversely, only 2.27 ± 0.55% of Syndecan3 objects co-localized with Synaptophysin. In addition, almost no immunostaining was observed in the stratum pyramidale of the CA1 area (Figure 6A–D). These results indicate that Syndecan3 is absent or expressed only at a very low level in CA1 pyramidal neurons or in perisomatic inhibitory synapses targeting CA1 pyramidal cell somas. Taken together with the relatively low affinity of GDNF to Syndecan3, these data suggest that Syndecan3 is not a major pathway responsible for the postsynaptic effect of GDNF leading to increased clustering of GABA_A_ receptors in the CA1 pyramidal neurons.

### 2.5. Potential Involvement of NCAM Pathway in GDNF Effect

To examine the contribution of the NCAM pathway, we used PP2, an antagonist of the Fyn kinase phosphorylation [28]. Since activated NCAM signals via Fyn phosphorylation, PP2 is expected to prevent the reduction of IPSC IEI and increased amplitude seen after GDNF exposure if NCAM is involved in the observed effect on inhibitory currents. However, we found no reversal of the GDNF effect when PP2 was added to the incubation medium (Figure 7). In fact, there was an opposite effect of PP2, decreasing the IEI and increasing the amplitude of IPSCs as analyzed by cumulative probability curves (Figure 7A, right panels), indicating that it was unlikely that the NCAM pathway was involved in the GDNF effect. 

We confirmed this by comparing ratios of phosphorylated Fyn to total Fyn, which is the downstream signaling pathway of NCAM activation. We observed no statistically significant difference between the groups treated with GDNF (0.966 ± 0.036, n = 4) or GDNF together with PP2 (0.876 ± 0.041, n = 4) as compared to controls (Figure 7B), indicating that at least in our conditions, phosphorylation of Fyn by GDNF-NCAM is not increased after GDNF incubation.

### 2.6. Validation in Human Brain Tissue

Next, considering the possible value in treating drug-resistant epilepsies with GDNF gene therapy, we asked whether the effect of GDNF on inhibitory transmission would also be present in the human epileptic brain. To address this question, we performed GDNF exposure experiments in human epileptic hippocampal slices from patients with drug-resistant temporal lobe epilepsy that underwent temporal lobe resection for therapeutic purposes. Assessing cumulative probability curves for sIPSCs, we observed, similar to rodent slices, a significant decrease in IEI in CA1-pyramidal neurons of GDNF-incubated hippocampal slices compared to control aCSF-incubated ones (Figure 8B). The sIPSC amplitude distributions showed, however, a slight decrease in GDNF-treated as compared to aCSF-treated control slices (Figure 8C). 

## 3. Discussion

Here, we demonstrate that elevated extracellular levels of GDNF in hippocampal slices result in increased inhibitory drive onto the pyramidal neurons of the CA1 area. This enhanced inhibition was especially apparent for the perisomatic area of the principal neurons. We also demonstrate an increase in GABA_A_ receptors’ scaffolding protein gephyrin immunoreactivity. Moreover, based on results from various experimental approaches, we propose that these alterations are mediated by Ret pathway activation by GDNF in combination with GFRα1.

### 3.1. Enhanced Synaptic Inhibition of Principal Neurons by Increased Extracellular Levels of GDNF

Previous studies from our and other groups have shown an anti-seizure effect of GDNF when either delivered or over-expressed in epileptic tissue in animal models of epilepsy [15,16,17]. However, there was a gap in the understanding of the mechanisms of GDNF action. With this study, we addressed which of the different GDNF signaling mechanisms might contribute to its seizure-suppressant effects.

It has been shown that GDNF can promote functional and morphological differentiation of GABAergic neurons via GFRα1 [29,30]. Moreover, the addition of soluble GFRα1 promoted GABAergic differentiation in vitro even in cells lacking Ret and NCAM [30], most likely acting via an additional receptor partner, later identified as Syndecan-3, found to be expressed on GABAergic interneurons [5]. In line with these observations, compromise in inhibitory neurons due to defects in GFRα1 signaling in cortical areas was shown to increase excitability and sensitivity to sub-threshold doses of epileptogenic agents [31]. Thus, one could hypothesize that increased extracellular levels of GDNF acting on GFRα1 and/or indirectly on Syndecan-3 would be able to ameliorate deficits of inhibitory neurotransmission in epileptic animals by supporting the survival of GABAergic neurons and perhaps even promoting inhibitory synaptogenesis by LICAM mechanism [4]. Here, we demonstrate that, indeed, incubation of mouse hippocampal slices with GDNF does enhance inhibitory drive onto the principal neurons, both by increased frequency and amplitude of IPSCs. Such simultaneous changes in both frequency and amplitude of synaptic events are commonly interpreted as alterations at the pre-synaptic site [26], although postsynaptic mechanisms cannot be excluded.

Interestingly, we have observed that GDNF-induced alterations in IPSCs in the CA1 pyramidal cells were associated with an increased proportion of high amplitude/fast rise time events. This was taken to suggest that there was a preferential increase in inhibitory postsynaptic events in the perisomatic area of principal neurons. Since the whole-cell patch pipettes are always attached to the cell soma, the IPSCs generated on remote dendritic branches are subjected to a filtering effect, according to cable theory [22], resulting in slower and lower amplitude events recorded at the pipette location. A similar distinction between fast and slow IPSCs has been used elsewhere to distinguish between perisomatic and dendritic inhibitory synapses [32]. The reason for the localization-specific effect of GDNF on inhibitory synapses remains unclear. Differential localization of RET in neuronal compartments is one possible explanation. Predominantly somatic expression of a specific RET isoform has been reported in, e.g., olfactory bulb neurons [33]. These isoforms are also subjected to differential trafficking in neurons [34]. Another explanation could be the changes in GABA_A_ receptor subunit composition. It has been shown that α1β2 and α3β2 subunit compositions exhibit different rise times (10–90%) [35]. However, these changes in subunit composition require longer than the 1 h incubation time used in our study. Yet another explanation is possible changes in access resistance. However, there were no significant differences in access resistance between neurons recorded from slices exposed to GDNF and control solutions.

Although the increased frequency of mIPSCs indicates a pre-synaptic site of action, we also observed an increased number of gephyrin immunoreactive puncta closely associated with the cell soma membranes of CA1 pyramidal neurons in GDNF-exposed slices. As gephyrin is a crucial protein for the postsynaptic expression of active GABA_A_ receptors, an increase in its immunoreactivity is commonly interpreted as increased clustering of GABA_A_ receptors [36]. In addition, we did not observe GDNF-induced changes in pre-synaptic release probability or releasable pool from GABAergic PV perisomatic terminals when selectively stimulated by optogenetics. While these data do not exclude the possibility that other interneuron subtypes could be affected differently by GDNF, they at least indicate that the direct pre-synaptic effect on PV terminals is not the main reason behind the observed increase in mIPSC frequency. 

Thus, a combination of our electrophysiological and morphological findings supports the idea that GDNF induces changes in GABA_A_ receptor efficacy, mostly at the postsynaptic sites of inhibitory synapses on CA1 pyramidal neurons. We did not observe any statistically significant increase in the numbers of GAD67 or PV interneuron terminals in the CA1 pyramidal layer. This finding indicates that an increase in GABA_A_ receptor clustering occurs in a subset of existing inhibitory synapses but not due to synaptogenesis.

The increased inhibitory synaptic drive was also observed in human hippocampal slices. The shift in the distribution of inter-event intervals was consistent with the mouse data and showed a higher frequency of sIPSCs after GDNF incubation, suggesting that GDNF may also have a seizure-suppressant effect in the human epileptic hippocampus, providing translational validation of the findings in rodents. However, the distribution of sIPSCs amplitudes after GDNF incubation in human slices was the opposite of what was observed in mice. The reason behind this discrepancy is currently unknown, but we have previously observed differences in the effects of neuropeptides between rodent and epileptic human tissue [37]. While it is possible that GDNF mechanisms might differ or be more complex in the human slices, it is also worth noting that human tissue is derived from patients with epilepsy, which, in addition, have been exposed to anti-seizure medications for several years, while mouse slices were from naïve animals. 

### 3.2. GDNF Signalling Pathways

Which receptors and downstream intracellular pathways are responsible for the observed GDNF effects? Based on the current literature, GDNF may exert an effect through Ret [2], NCAM [3], or Syndecan-3 [5] receptors (see Figure 8). The series of experiments performed by us suggests that the Ret-dependent pathway is the most important one. The basis for this conclusion is provided by several experiments. First, a positive modulator of the GFRα1-Ret pathway (XIB4035) further enhanced the GDNF-induced increase in IPSC frequencies and amplitudes. Second, GDNF incubation increased Ret phosphorylation, suggesting its activation. Third, the GDNF-induced phosphorylation of Ret was reversed by the Ret inhibitor SPP86.

The involvement of other receptors and pathways that are activated by GDNF is less obvious. The Fyn phosphorylation, indicative of activation of NCAM downstream pathway, was not affected by GDNF exposure. Moreover, since in CA1 pyramidal neurons, we could not find any evidence of Syndecan3 expression, its involvement in GDNF-induced increase in GABAA receptor clustering can be excluded. In addition, Syndecan3 has a lower affinity to GDNF compared to, e.g., Ret [5,38].

Taken together, all these data suggest that the activation of the GDNF-GFRα1-Ret pathway is the putative molecular mechanism of enhanced synaptic inhibition. This effect of GDNF is achieved by increasing the number of clustered GABA_A_ receptors at postsynaptic sites, predominantly at the perisomatic area of the principal neurons (Figure 9).

These findings provide at least partial clarification of the previously anti-seizure effects of GDNF observed in various acute and chronic models of epilepsy [15,16,17]. Whether the concentration of GDNF used here can be achieved in vivo needs to be established. In the previous in vivo studies with encapsulated GDNF-releasing cells, the concentration of GDNF reached approximately 4 nM [15], which is 2–3 orders of magnitude lower than that used by us. Follow-up studies need to address this issue in more detail. 

There are several additional hypothetical mechanisms for how increased extracellular levels of GDNF may suppress seizures in models of chronic epilepsy. It has been shown that blood–brain barrier (BBB) disruption allows blood albumin to penetrate the brain and thereby induce excitatory synaptogenesis, leading to increased excitability and seizures [39]. Since GDNF was found to increase the expression of Claudin-5 [40], one of the most important molecules maintaining BBB integrity. Thus, the compromised BBB in chronic epilepsy can be reinstated by a GDNF-mediated increase in Claudin-5 and thereby decrease counteract seizures. Another potential hypothetical mechanism is the regulation of inflammation associated with the chronic epileptic state [41]. It has been shown that GDNF released from astrocytes reduces the production of reactive oxygen species and phagocytosis by activated microglia [21], both associated with inflammation. Thus, counteracting the brain inflammation may also lead to decreased occurrence of chronic seizures.

In conclusion, here we identified a previously unknown mechanism of GDNF action enhancing inhibitory drive onto the hippocampal principal neurons. This novel mechanism can explain, at least partially, the seizure-suppressant effects of GDNF observed earlier in animal models. These findings will also stimulate further research, ultimately leading to the development of GDNF-based therapies against epilepsy. One might envisage a future clinical application whereby overexpression of GDNF together with related receptors would enhance the inhibition of principal neurons in the hippocampus and thereby counteract focal seizures.

## 4. Materials and Methods

### 4.1. Code Accessibility

The whole original code has been deposited at GitHub and Zenodo and is publicly available as of the date of publication at https://github.com/AMikroulis/xPSC-detection accessed on 25 October 2022(xPSC-detection-Template correlation-based detection of postsynaptic currents, https://github.com/AMikroulis/staining-analysis/ accessed on 9 October 2022).

Any additional information required to reanalyze the data reported in this paper is available from the lead contact upon request.

### 4.2. Animals and Ethical Information

Mice with C57/BL6 background from Jackson Laboratory were bred and kept at stables in standard cages with ad libitum access to food and water and a 12 h light/dark cycle. Mouse experiments were conducted in compliance with Swedish law under ethical permit number 02998/2020.

The use of resected patient tissue from Lund University Hospital and Copenhagen University Hospital, Rigshospitalet, and the following procedures were approved by the local Ethical Committee in Lund (#212/2007) and Copenhagen (H-2-2011-104), respectively. All experiments were performed in accordance with the Declaration of Helsinki. Written informed consent was obtained from all subjects prior to each surgery (age and sex of patients is reported on Table 3).

### 4.3. Slice Preparation

Briefly, 3–10-week-old mice were anesthetized with isoflurane and sacrificed, and the brain was quickly removed and subsequently processed in ice-cold cutting aCSF (75 mM sucrose (VWR, Radnor, PA, USA), 66.9 mM NaCl (VWR), 2.5 mM KCl (Merck, Kenilworth, NJ, USA), 0.5 mM CaCl_2_ (Merck), 7 mM MgCl_2_ (Sigma-Aldrich, St. Louis, MO, USA), 1.25 mM NaH_2_PO_4_ (Merck), 26 mM NaHCO_3_ (Merck), and 25 mM D-glucose (VWR)). The cerebellum was removed, and 300 µm horizontal slices were cut with a Leica 1200S VT vibratome, starting from the ventral part of the brain.

From two to three slices were kept for electrophysiology on the condition that they contained the ventral hippocampus, typically between 900 µm and 1800 µm from the ventral end of the brain. The slices (containing the posterior part of both hemispheres) were further dissected into 2 smaller slices of the hippocampus and peri-hippocampal areas, with a sagittal plane cut medially to the outer shell of the dentate gyrus. This was conducted to enable easier transport between chambers, reduce the glucose and oxygen demands and diffused metabolite accumulation during incubation in the limited volume chamber, and reduce the debris in circulation in the incubation and recording chamber.

For the human tissue experiment, epileptic human hippocampal slices were cut and maintained from tissue received from surgical resections performed at Lund University Hospital and Rigshospitalet University Hospital, as previously described [42,43]. Briefly, resected tissue was collected in an ice-cold sucrose-based cutting solution containing (in mM): 200 sucrose, 21 NaHCO_3_, 10 glucose, 3 KCl, 1.25 NaH_2_PO_4_, 1.6 CaCl_2_, 2 MgCl_2_, 2 MgSO_4_ (all from Sigma-Aldrich), adjusted to 300–310 mOsm, 7.4 pH. The 300 µm slices were cut with a vibratome (Leica VT1200S, Wetzlar, Germany) and transferred to a submerged incubation chamber filled with aCSF, containing (in mM): 129 NaCl, 21 NaHCO_3_, 10 glucose, 3 KCl, 1.25 NaH_2_PO_4_, 2 MgSO_4_, and 1.6 CaCl_2_, adjusted to 300–310 mOsm, 7.4 pH, heated to 34 °C and continuously bubbled with carbogen (95% O_2_ and 5% CO_2_). Slices rested submerged for 15–30 min before being transferred to cell culture membranes inside a humidified interface holding chamber containing the same aCSF, where they were maintained for at least 24 h before the start of the recordings [43].

### 4.4. Slice Incubation

From one to two slices per animal per condition were incubated at room temperature for 1 h in a custom-designed low-volume (20 mL) chamber with freely circulating aCSF (118 mM NaCl, 2.5 mM KCl, 1.25 mM NaH_2_PO_4_, 2 mM MgCl_2_, 2 mM CaCl_2_, 10 mM D-glucose) or aCSF with 2 nM of mouse GDNF (Sigma-Aldrich). The aCSF for human slice recording contained 129 mM NaCl, 3 mM KCl, 21 mM NaHCO_3_, 10 mM D-glucose, 2 mM MgSO_4_, 1.6 mM CaCl_2_ and 1.25 mM NaH_2_PO_4_. Alternatively, the slices were incubated with 10 µM XIB4035 (Sigma-Aldrich), 1 µM PP2 (Tocris, Bristol, UK), 0.1% DMSO control aCSF, GDNF 2 nM with 0.1% DMSO, or a combination of the above (Figure 10). The human hippocampal slices were moved from the interface holding chamber to the low volume (20 mL) chamber with either aCSF or aCSF with GDNF (2 nM) for 1 h before recording.

### 4.5. Patch-Clamp Recordings

Glass pipettes were pulled using thick-wall Stoelting (ID/OD 0.75/1.50) or King Precision borosilicate glass (ID/OD 0.86/1.50) on a Sutter P-97 puller and filled with a pipette solution containing CsCl 135 mM, NaCl 8 mM, CsOH EGTA 0.2 mM, CsOH HEPES 10 mM, MgATP 2 mM, Na_3_GTP 0.3 mM, QX-314 5 mM and 0.2% biocytin. Pipette resistance was in the range of 2–5 MOhm.

Spontaneous and miniature (tetrodotoxin citrate 1 µM) postsynaptic currents were recorded at 10 kHz sampling rate, after a 3 kHz antialiasing Bessel filter, in gap-free mode from mid-distal CA1 pyramidal neurons, voltage-clamped at –70 mV, with either NBQX 5 µM and D-AP5 50 µM or PTX 100 µM. A HEKA EPC9 and HEKA Patchmaster v13.52 for Apple-macOS was used for acquisition.

For the light-evoked IPSC recordings, slices from seven PV-ChR2 mice were incubated for 1h with either aCSF or aCSF with 2 nM GDNF. One cell was recorded per slice, and at least one control slice and one incubated slice were used from each mouse. Similar to the previous experiment, whole-cell patch-clamp recordings from CA1 pyramidal neurons were used to record responses from 12 trains, 15 s apart, consisting of 20 light pulses (3 ms duration each) at 20 Hz, delivered through the microscope objective from a 460 nm LED light source (Prizmatix, Holon, Israel), set at 60% output power. The responses were sampled at 20 kHz, with a 3 kHz antialiasing (Bessel low-pass) filter.

### 4.6. Immunohistochemistry–Imaging

Recorded slices were processed for biocytin. Biocytin-streptavidin staining consisted of 3 10-minute washes in KPBS, followed by 2 30-minute washes in 0.25% Triton-x100 KPBS, 3 hour incubation at room temperature with 1:2000 streptavidin-Cy5 (Jackson Immunoresearch 016-170-084) in Triton-KPBS, 3 20-minute washes in KPBS and mounting with DABCO. 

Epifluorescence images were taken to morphologically confirm stained pyramidal neurons. For a subset of the recorded slices, subslicing to 30 µm sections was performed with a microtome to stain for gephyrin, GAD65/67, and parvalbumin. 

Parvalbumin staining consisted of 3 10-minute washes in KPBS, 1 h blocking with 10% donkey serum in KPBS, 1:1000 mouse parvalbumin primary antibody (Swant 235) with 5% donkey serum in 0.25% Triton-x100 KPBS overnight incubation at 4C, 3 10-minute washes in KPBS, 2h incubation with secondary + 1% donkey serum, and rinsing thrice with KPBS.

Gephyrin staining (Synaptic Systems 147,011, Rabbit anti-gephyrin antibody) required an additional antigen-retrieval step with citric buffer (10 mM sodium citrate, 0.05% Tween 20) at 90 C for 20 min, and subsequent staining with 1:500 gephyrin primary antibody. The remaining steps were performed in tandem with the parvalbumin staining.

GAD65/67 staining consisted of 3 10-minute washes in KPBS, 1h blocking with 10% donkey serum in KPBS, 1:1000 rabbit GAD65/67 primary antibody (Sigma-Aldrich G5163) with 5% donkey serum in 0.25% Triton-x100 KPBS overnight incubation at 4 C, 3 10-minute washes in KPBS, 2 h incubation with secondary + 1% donkey serum, and rinsing thrice with KPBS.

For all fluorescence stainings, the secondary antibodies were added after 3 × 10 min rinses in KPBS, followed by 2 h incubation at room temperature with 1:500 Alexa Fluor 488 (Thermo Fisher, Waltham, MA, USA) or Alexa fluor PLUS 555 (Invitrogen, Waltham, MA, USA), or Cy3 or Cy5 fluorophore-conjugated antibodies (Jackson immunoresearch, West Grove, PA, USA) depending on the primaries. Nuclei were stained using DAPI/Hoescht 1:1000 in the mounting medium (DABCO).

Quantification of gephyrin, PV, and GAD65/67 staining was conducted on confocal images of single 93 × 93 µm (948 × 948 px, 0.09822 µm/px) CA1 pyramidal fields (one field per slice, one slice per animal, repeated for 7 animals for gephyrin and PV staining, and for 5 animals for GAD65/67 staining). The images were acquired with identical settings. The analysis was conducted by >5 SD brightness-based point counting after a median filter for background and acquisition noise filtering, respectively. Cell nuclei were identified using the overlap of the Chan-Vese/watershed segmentation algorithm output and a Gaussian-kernel Laplace filter to exclude frame edge false positives. Distance of staining points from cell nuclei was taken into account in the counting process: staining points were excluded within a 5 µm radius from the detected nuclei centroids to limit intracellular staining (the entire analysis procedure is in https://github.com/AMikroulis/staining-analysis accessed on 25 October 2022). The procedure was repeated in all processed images.

### 4.7. Array Tomography

Hippocampi were dissected from C57BL/6 mice and prepared for array tomography as outlined previously [44]. Briefly, fresh post-mortem mouse brains were dissected, and the hippocampus was trimmed into blocks and fixed in 4% paraformaldehyde in PBS for 2–3 h. Samples were then dehydrated through ascending ethanol washes (50%, 70%, 90%, and 100%) and into LR White resin overnight. Blocks were placed individually into capsules containing LR White and polymerized overnight at 60 °C. Tissue blocks were cut into ribbons of serial sections of 70 nm thickness using a Leica UC7 microtome with a Histo Jumbo Diamond knife (Diatome, Hatfield, PA, USA) and collected on gelatin-coated glass coverslips. Ribbons were treated with 50 mM glycine for 5 min and blocked for 30 min (0.1% fish skin gelatin and 0.05% Tween20 in TBS). Afterward, they were immunostained as described previously [45] with primary antibodies against synaptophysin (1:50, ab8049, Abcam, Cambridge, UK) and syndecan 3 (1:50, 10886-1-AP, Proteintech, Rosemont, IL, USA) overnight at 4 °C. No-primary negative control was included to rule out non-specific binding of the secondary antibodies. The following day, the staining was developed with fluorescently labeled secondary antibodies (1:50 donkey anti-mouse Alexa fluor 488–ab150105, Abcam; 1:50 donkey anti-rabbit Alexa fluor 594–ab150076, Abcam, respectively) for 1 hr and mounted onto slides with Immumount mounting media. Once the ribbons had been stained and mounted, images were obtained at the same position on each section along the ribbon using a DeltaVision Elite widefield fluorescence microscope (Image solutions) equipped with a CoolSnap digital camera and softWoRx software. High-resolution images were obtained with a 63X 1.4NA Plan Apochromat objective. At least two image stacks were captured per block from each mouse.

Stacks were aligned using the ImageJ Multistack Reg plugin [46]. After thresholding, a Watershed script [47] was used to remove false background staining found only on single sections. Finally, the ImageJ 3D viewer tool was used to generate 3D reconstructions of the images.

### 4.8. Western Blot

A subset of the incubated slices was used for phosphorylated Ret and Fyn quantification by Western blot. Samples were collected after incubation into cold N-PER Neuronal Protein Extraction Reagent containing Halt Protease and Phosphatase Inhibitor Cocktail (Thermo Scientific); they were directly homogenized and kept on ice for 10 min. Samples were then centrifuged for 10 min at 10.000× *g* at 4 °C. Supernatants were collected, aliquoted, and stored at −80 °C until further use. 

Primary antibodies and their used concentrations were: Recombinant Anti-Ret Antibody (Abcam, ab134100) 1:200; Anti-phospho-Ret (pTyr1062) antibody (Sigma-Aldrich, SAB4504530) 1:200; Fyn Antibody (#4023, Cell Signaling Technology) 1:500; Anti-Fyn (phospho Y530) antibody (ab182661, Abcam) 1:500. 

Total protein concentrations were determined by Pierce BCA Protein Assay Kit (Thermo Scientific) according to the manufacturer’s directions. Samples were denatured at 95 °C for 5 min in Bolt LDS Sample Buffer containing Bolt Sample Reducing Agent (Invitrogen). In total, 30 ug of protein per sample was subjected to SDS-PAGE on Bolt 4-12% Bis-Tris Gels with Bolt MES SDS Running Buffer (Invitrogen) and PageRuler Plus Prestained Protein Ladder (Thermo Scientific). After electrophoresis, gels were blotted onto PVDF membranes using Trans-Blot Turbo Mini PVDF Transfer Packs and the Trans-Blot Turbo Transfer System (Bio-Rad, Hercules, CA, USA) on a High MW program. The transferred membranes were directly incubated in 0.4% paraformaldehyde (PFA) in PBS for 30 min at RT and then thoroughly rinsed with water. Membranes were then washed with TBS containing 0.1% (*v*/*v*) Tween 20 (TTBS), blocked in TTBS containing 5% skim milk for 90 min at RT, and further incubated in TTBS containing 1% Bovine Serum Albumin (BSA) with diluted primary antibodies against phosphorylated proteins and non-phosphorylated proteins (where molecular weight differed) overnight at 4 °C. The next day membranes were thoroughly washed in TTBS. They were incubated in a blocking solution containing the corresponding HRP-coupled secondary antibodies and Anti-beta Actin antibody (HRP-coupled) (Abcam, ab49,900) for 90 min at RT, and subsequently washed in TTBS and TBS. To visualize the signal, membranes were exposed to SuperSignal West Pico PLUS Chemiluminescent Substrate (Thermo Scientific). Signals were detected using a ChemiDoc Imaging System (Bio-Rad). After detection, membranes were stripped using Restore PLUS Western Blot Stripping Buffer (Thermo Scientific) for 15 min and then briefly washed with TBS, blocked, and re-stained with total protein antibodies. Levels of total and phosphorylated proteins were estimated by measuring band intensities with Image Lab Software (Bio-Rad). For Phospho-Ret and Ret analysis, the combination of the two bands was considered as total Ret; for Phospho-Fyn and Fyn analysis, the Fyn band was considered as total Fyn due to the identical molecular weight. In all cases, beta-actin was used as a loading control.

### 4.9. Quantification and Statistical Analysis

Postsynaptic currents were detected with the correlation coefficient method [48] and analyzed using Intel Python 3.6 under Windows 10 (code available at [49]). Events were selected if their correlation coefficient with a double exponential fit of averaged manually identified postsynaptic currents exceeded 0.6, with an amplitude greater than 3 pA (amplifier noise floor), 20–80% rise time less than 5 ms and halfwidth greater than the 20–80% rise time, to exclude potential artifacts. In total, 20–80% rise times were reported throughout.

The fast and slow rise-time events were separated with a discrete method [50]:

For all possible segmentation points in the rise-time values, *τ_c_* in δτ intervals, *δ* denoting the minimum discrete difference,
(1)τc=k·δτ, k ∈ ℕ, k>0,δτ>0
we calculate the ratio, *r_s:f_*, of counts of observed rise-times slower-than to faster -than *τ_c_*,
(2)$$r_{s:f}=\frac{n_{\tau >\tau_{c}}+1}{n_{\tau \leq \tau_{c}}+1}$$
and select the minimum *τ_c_* where the rates of change of $r_{s:f}$ for the 2 groups are equal or approximately equal after the r_s:f_ spike at the beginning of the *τ_c_* range:(3)$$\hat{\tau_{c}}=min\left(\tau_{c}\right):r_{s:f}\;\ll \;max\left(r_{s:f}\right),\frac{\delta r_{s:f,ctrl}}{\delta \tau }\;≅\;\frac{\delta r_{s:f,GDNF}}{\delta \tau }$$

*δτ* = 100 µs (sampling rate limit).

To analyze the releasable pool, the light pulse trains (8 ctrl and 8 GDNF slices from 7 animals) were analyzed following the SMN-fit method described in [51] after averaging the traces and using the last 3 pulse time-points for the fit region. To account for cell-to-cell and slice-to-slice variance in afferent PV innervation, the amplitude intercepts were scaled to the amplitude of the first light pulse response. The release probability (7 ctrl and 7 GDNF slices from 6 animals) was estimated following the Bayesian estimation method described in [52] using the individual light pulse trains before averaging and by repeating the analysis 10 times and averaging the results to reduce parameter space sampling error. Briefly, the algorithm fits a binomial model (for a single synapse basis) to the data (evoked IPSC amplitudes). The parameters for the model (release probability, number of release sites, quantal size, and variances) are evaluated starting with a random point on a five-dimensional grid and scanning nearby parameter values until the likelihood function is maximized. At that point, the parameters are aggregated. The process is repeated 10 times to reduce the parameter sampling noise, and the aggregated parameters are averaged first on a per-cell basis and then on a per-treatment (Control or GDNF) basis.

The Kolmogorov–Smirnov test was used to compare cumulative probability distributions (accepted as statistically significant differences with *p* < 0.01 and D > 0.05). The same number of consecutive events per cell in each condition was used for even weighed contribution of each cell to the group distributions. The Mann–Whitney U test was used to compare independent sample pairs (*p* < 0.05). Differences were inferred on the cell/slice level from the Mann–Whitney U test and the Kolmogorov–Smirnov test. A total of 29 animals were used for electrophysiology (7 animals for ctrl-GDNF IPSCs, 5 animals for ctrl-GDNF EPSCs, and 7, 4, and 6 animals for the XIB4035, SPP86 (Extended Data) and PP2 experiments, respectively) and gephyrin/PV/GAD65-67 staining, 3 animals were used for perfusion/confocal/TEM imaging, 4 animals were used for immunoblots. No more than 2 slices were used per animal per incubation category for electrophysiology, 1 slice per animal was imaged for gephyrin immunohistochemistry. The number of cells is reported as *n* for all electrophysiology measurements, with one cell used per slice. The Friedman test with Dunn’s post-hoc test was used for multiple paired comparisons (*p* < 0.05). Fisher’s exact test was used for proportion comparisons (*p* < 0.05). In the Western blot experiments, for comparison between conditions, the Friedman test with Dunn’s multiple comparisons test was used. 

Statistics were performed in Python 3 (Intel), Statistica 13 (TIBCO), or Prism 8 (Graphpad).

## Figures and Tables

**Figure 1 ijms-23-13190-f001:**
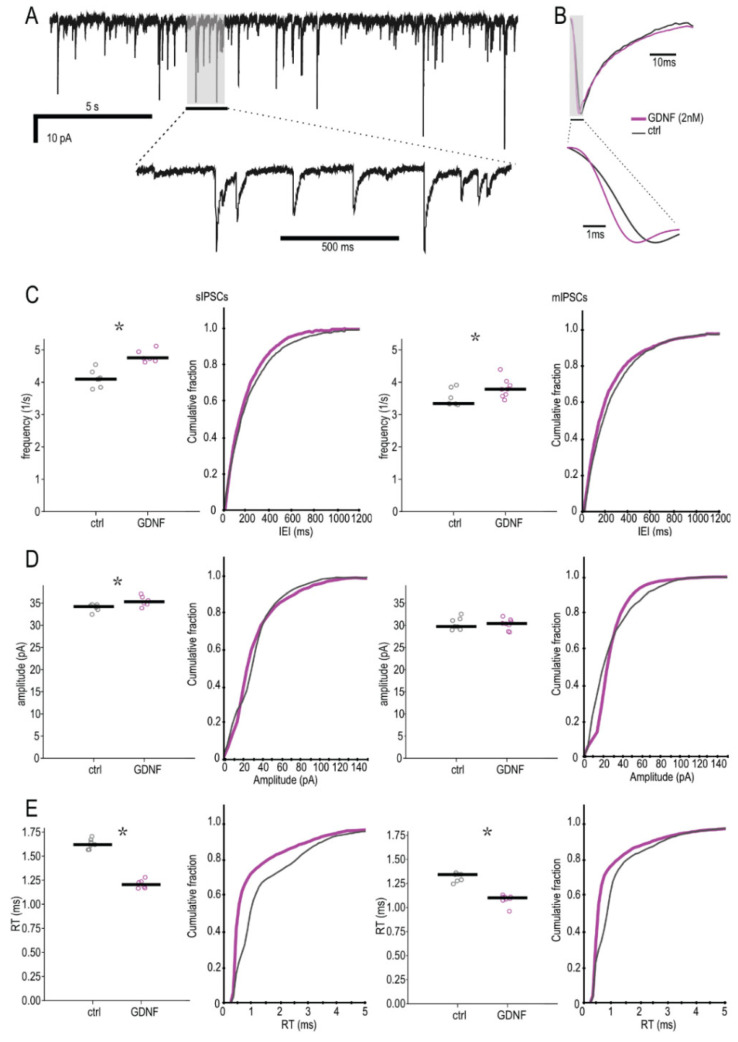
Quantification of GDNF effect on inhibitory postsynaptic currents. (**A**) Example trace of spontaneous IPSCs from a control slice. (**B**) Comparison of the averaged sIPSCs from a control and a GDNF-incubated slice from the same animal, highlighting the difference in rise times (normalized amplitudes). (**C**–**E**) Inter-event interval (K-S sIPSCs *p* < 0.01 D = 0.238, mIPSCs *p* < 0.01 D = 0.075), amplitude (K-S sIPSCs *p* < 0.01 D = 0.127, mIPSCs *p* < 0.01 D = 0.213) and rise-time (K-S sIPSCs *p* < 0.01 D = 0.350, mIPSCs *p* < 0.01 D = 0.309) cumulative distribution plots of spontaneous (**left**) miniature and IPSCs (**right**) from control and GDNF-incubated slices (n = 352 events per cell). The line markers in the scatter plots depict the median of averages per cell. Mann–Whitney U-test for the averages. * *p* < 0.05.

**Figure 2 ijms-23-13190-f002:**
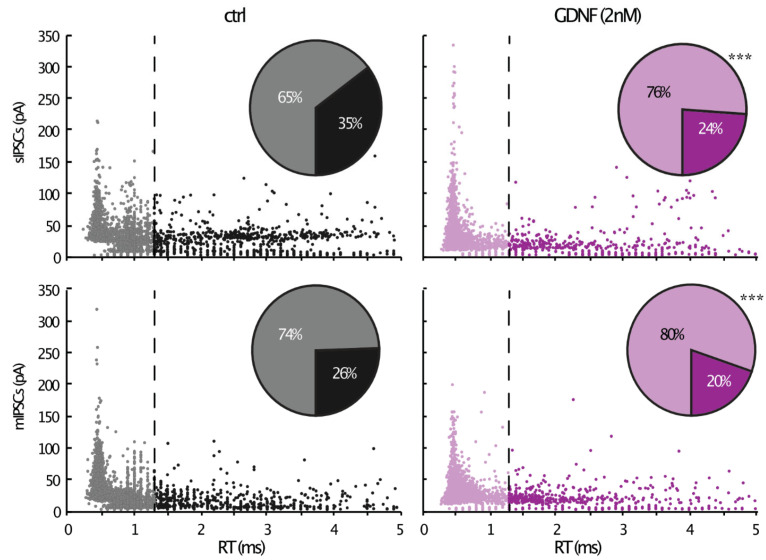
Bivariate plot of amplitude and rise time for each event. The pie charts depict the percentage of fast and slow events. ***: Fisher’s exact test *p* < 0.001.

**Figure 3 ijms-23-13190-f003:**
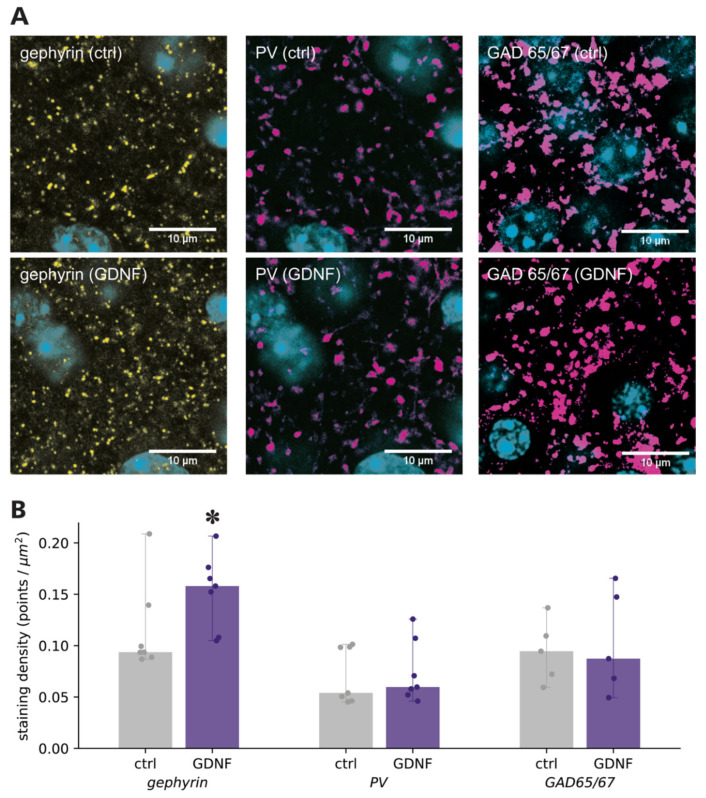
Confocal microscopy of gephyrin and inhibitory interneurons. (**A**) Gephyrin (yellow), parvalbumin, and GAD65/67 staining (magenta) for control and GDNF-incubated slices (Hoescht staining in blue), and (**B**) quantification of their staining density * Mann–Whitney U-test *p* < 0.05.

**Figure 4 ijms-23-13190-f004:**
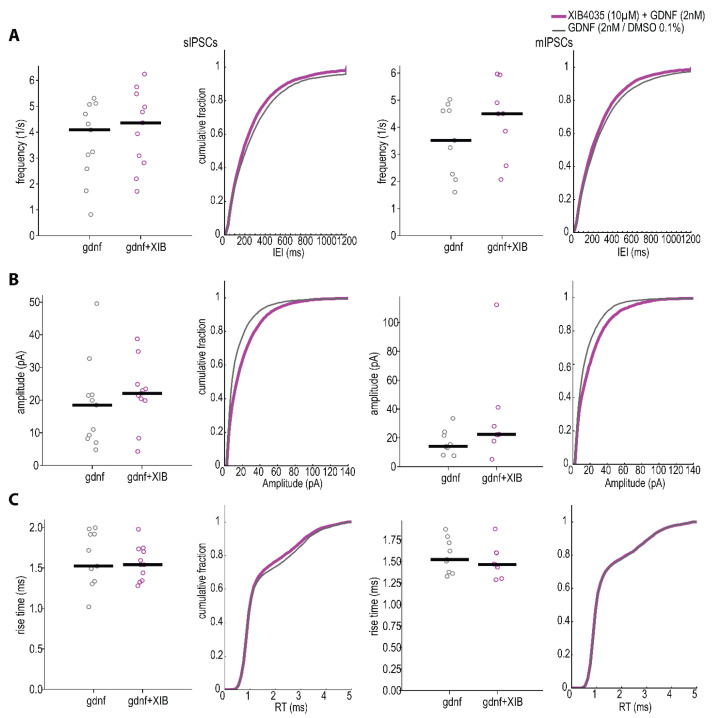
Quantification of Ret activation enhancement effect on inhibitory postsynaptic currents using XIB4035. (**A**) Inter-event interval (K-S sIPSCs *p* < 0.01 D = 0.068, mIPSCs *p* < 0.01 D = 0.088), (**B**) amplitude (K-S sIPSCs *p* < 0.01 D = 0.144, mIPSCs *p* < 0.01 D = 0.166) and (**C**) rise-time (K-S sIPSCs *p* < 0.01 D = 0.032; mIPSCs *p* < 0.01 D = 0.034) cumulative distribution plots of spontaneous (**left**) and miniature IPSCs (**right**) from control and GDNF-incubated slices (n = 511 events per cell). The line markers in the scatter plots depict the median of averages per cell. Mann–Whitney test for the averages–not significant, Kolmogorov–Smirnov test for distribution comparisons. *p* < 0.01.

**Figure 5 ijms-23-13190-f005:**
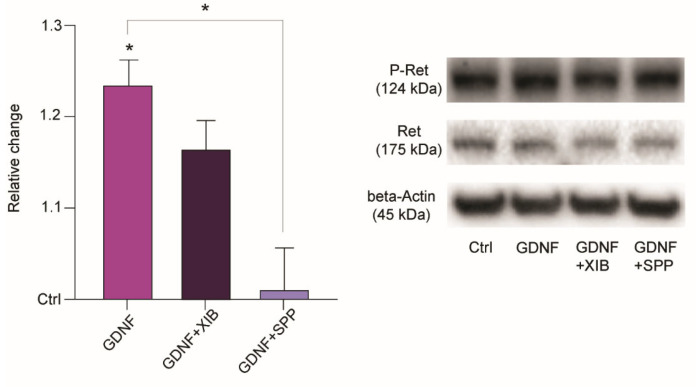
Relative change in phosphorylation ratio of Ret after treatment with GDNF, GDNF + XIB4035 and GDNF + SPP compared to control. Example images of Western blot membrane staining depicted to the right. Data represented as mean and SEM. Friedman test, Dunn’s multiple comparisons test-GDNF to control, * *p* < 0.05; GDNF + SPP to GDNF, * *p* < 0.05; n = 4 for each condition.

**Figure 6 ijms-23-13190-f006:**
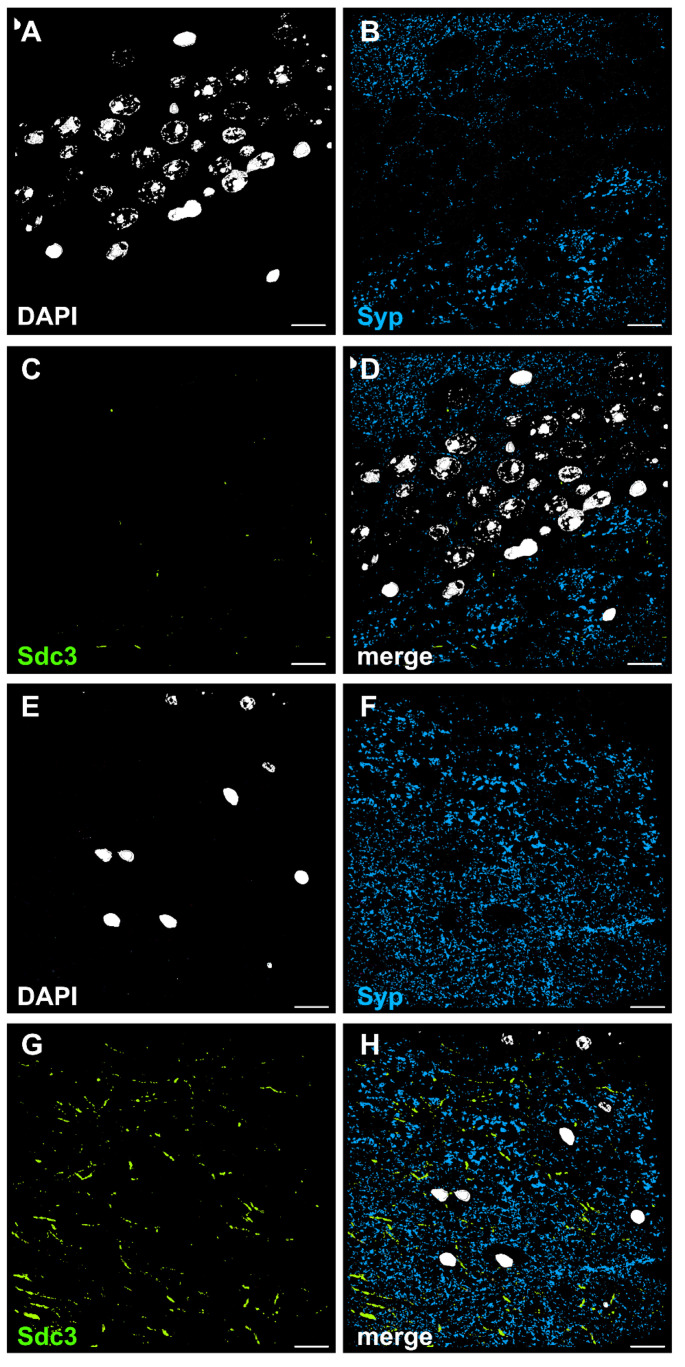
Array tomography images demonstrating localization of Syndecan3-immunoreactive spots in mouse hippocampus. (**A**–**D**) Images taken in CA1 stratum pyramidale, 3D render of 20 consecutive 70 nm sections. (**A**) DAPI, (**B**) Synaptophysin (Syp), (**C**) Syndecan3 (Sdc3), (**D**) merged. (**E**–**H**) Images taken in CA1 stratum radiatum, 3D render of 16 consecutive 70 nm sections (**E**) DAPI, (**F**) Synaptophysin, (**G**) Syndecan3, (**H**) merged. Scale bars: 50 µm.

**Figure 7 ijms-23-13190-f007:**
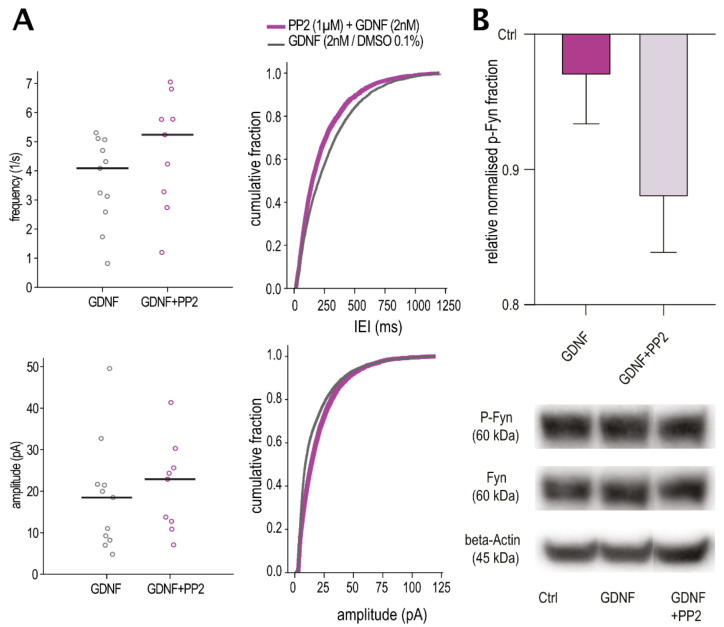
No effect of NCAM inhibition. (**A**) Inter-event interval and amplitude cumulative distribution plots of spontaneous IPSCs from GDNF and GDNF + PP2-incubated slices (n = 511 events per cell). The line markers in the scatter plots depict the median of averages per cell. Mann–Whitney test for the averages–not significant, Kolmogorov–Smirnov test for distribution comparisons (IEI: D = 0.126, amplitude: D = 0.155). (**B**) Relative change in phosphorylation ratio of Fyn after treatment with GDNF and GDNF + PP2 compared to control (n = 4). Example images of Western blot membrane staining depicted below. Data represented as mean and SEM. Friedman test—not significant.

**Figure 8 ijms-23-13190-f008:**
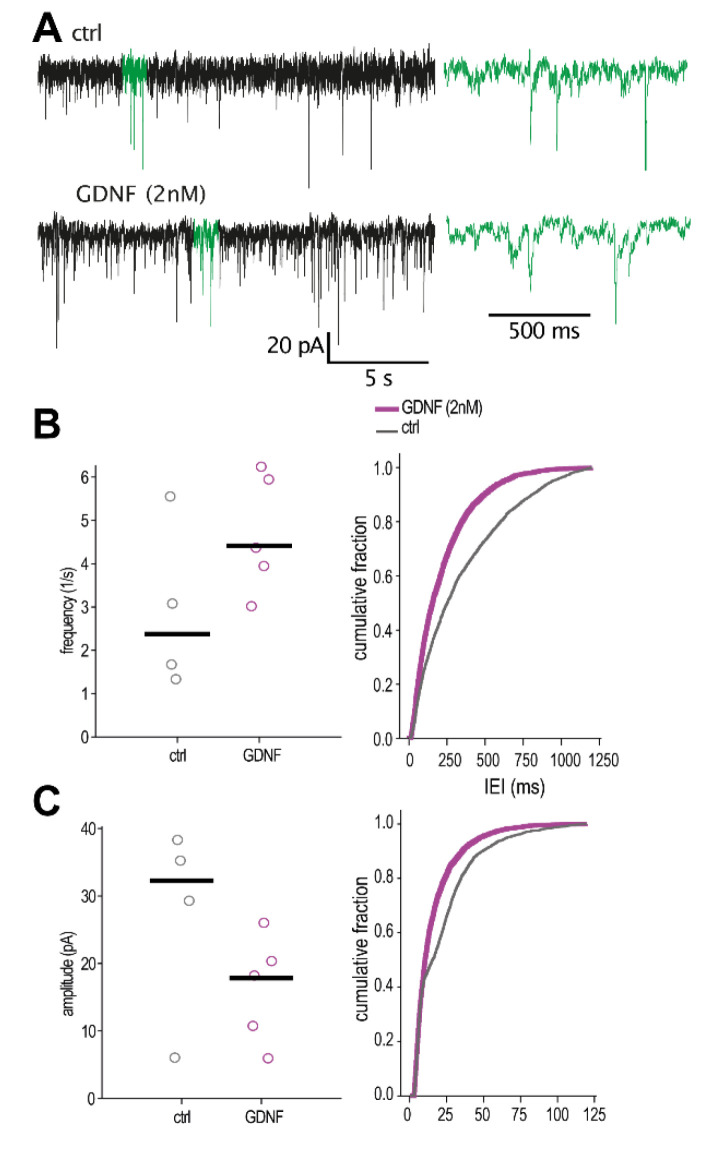
Inter-event interval and amplitude of spontaneous IPSCs from GDNF-incubated human epileptic hippocampal slices. (**A**) Representative traces of sIPSC recordings. Green-colored areas are magnified on the right. Quantification of (**B**) frequency and (**C**) amplitude of spontaneous IPSCs. The line markers in the scatter plots depict the median of averages per cell. Mann–Whitney U-test for the averages–not significant; n = 4 cells for controls and n = 5 cells for GDNF; Kolmogorov–Smirnov test for distribution comparisons, *p* < 0.01 (IEI: D = 0.238, amplitude: D = 0.199); n = 744 events per cell.

**Figure 9 ijms-23-13190-f009:**
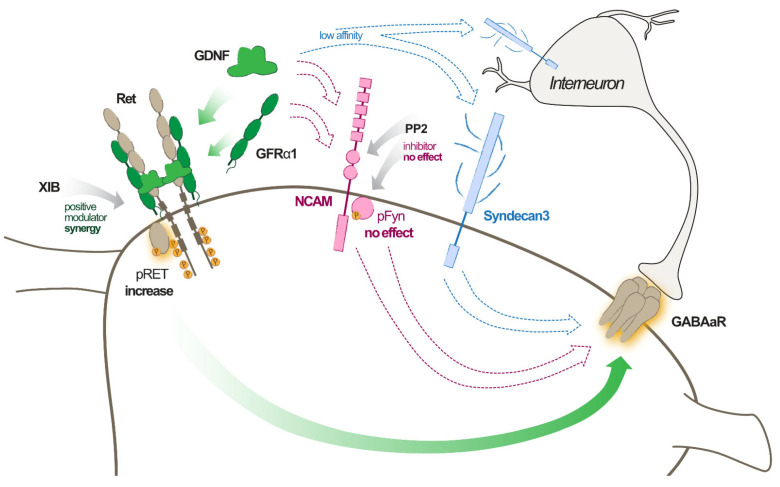
Graphical illustration of likely GDNF mechanism of action leading to increased GABA_A_R effect on pyramidal neurons in the hippocampus.

**Figure 10 ijms-23-13190-f010:**
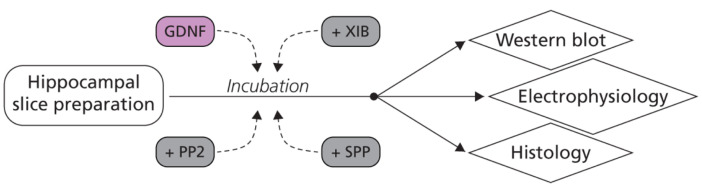
Graphical illustration of experimental methods. After slice preparation (from mouse or human tissue), slices were incubated with GDNF, XIB4035, PP2, SPP86, or combinations for 1 h before being processed for either Western blot, electrophysiology, or histology experiments.

**Table 1 ijms-23-13190-t001:** Comparison of IPSC averages ± standard error of mean (and median) by cell following control and GDNF incubation for 1 h. Frequency, amplitudes, and rise time values of IPSCs from control and GDNF incubated slices are shown along with the Mann–Whitney *p*-value. The number of cells recorded is presented as n.

	Frequency (Hz)		Amplitude (pA)		Rise Time (ms)	
	sIPSCs	mIPSCs	sIPSCs	mIPSCs	sIPSCs	mIPSCs
Ctrl	4.1 ± 0.1 (4.1), n = 7	3.5 ± 0.1 (3.4), n = 7	34.0 ± 0.3 (34.2), n = 7	30.3 ± 0.5 (29.7), n = 7	1.62 ± 0.01 (1.62), n = 7	1.31 ± 0.01 (1.34), n = 7
GDNF 2nM	4.8 ± 0.1 (4.8), n = 6	3.8 ± 0.1 (3.8), n = 8	35.4 ± 0.4 (35.3), n = 6	30.2 ± 0.4 (30.4), n = 8	1.21 ± 0.01 (1.20), n = 6	1.08 ± 0.01 (1.10), n = 8
Mann–Whitney *p*	0.002	0.036	0.013	0.477	0.002	0.001

**Table 2 ijms-23-13190-t002:** Comparison of IPSC averages ± standard error of mean (and median) by cell following GDNF in 0.1% DMSO and GDNF/XIB4035 incubation for 1h. Frequency, amplitudes, and rise time values of IPSCs from slices incubated with GDNF only or GDNF and XIB4035 are shown along with the Mann–Whitney *p*-value. The number of cells recorded is presented as n.

	Frequency (Hz)		Amplitude (pA)		Rise Time (ms)	
	sIPSCs	mIPSCs	sIPSCs	mIPSCs	sIPSCs	mIPSCs
GDNF 2nM (DMSO)	3.6 ± 0.4 (4.1), n = 11	3.5 ± 0.4 (3.5), n = 9	18.5 ± 3.8 (18.5), n = 11	16.8 ± 2.6 (14.0), n = 9	1.60 ± 0.09 (1.52), n = 11	1.56 ± 0.06 (1.52), n = 9
GDNF 2nM + XIB4035	4.1 ± 0.4 (4.4), n = 11	4.3 ± 0.4 (4.5), n = 8	21.9 ± 2.8 (22.1), n = 11	33.8 ± 11.0 (22.3), n = 8	1.56 ± 0.06 (1.54), n = 11	1.50 ± 0.063 (1.47), n = 8
Mann–Whitney *p*	0.277	0.180	0.119	0.056	0.396	0.235

**Table 3 ijms-23-13190-t003:** Patient information table.

Patient	Age (Years)	Sex
1	18	Male
2	4	Female
3	43	Female

## Data Availability

Data used for this study are available on request from the corresponding author.

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
