# Peer review of "GDNF Increases Inhibitory Synaptic Drive on Principal Neurons in the Hippocampus via Activation of the Ret Pathway"

_ijms, 2022, doi:10.3390/ijms232113190_

Round 1

Reviewer 1 Report (New Reviewer)

1) What the horizontal bars in Fig 1C~E and Fig 4?

There are not identical between the horizontal bars and the calculated values in Table 1 and 2 

For example, 

 Table 2: Frequency sIPSCs-GDNF 3.6±0.4

 Fig 4A: horizontal bar for sIPSCs-gdnf was over "4"

2) Figure 8

Authors conducted these experiments using tissue from 3 patients, who were quite different for age and sex, in Table 3.

Descriptions for "n=4 for controls and n=5 for GDNF" in footnote of Fig 8 confused many readers how select tissues from 3 patients. 

Furthermore, it sounds the individual difference between patients might be negligible for these experiments.

3) Reference styles have to recheck very carefully (Abbreviations, sometimes found "Is" etc)

Foe example " J Neurosci Official J Soc Neurosci" in ref 31 

31. Canty, A.J.; Dietze, J.; Harvey, M.; Enomoto, H.; Milbrandt, J.; Ibáñez, C.F. Regionalized Loss of Parvalbumin Interneurons in the 780 Cerebral Cortex of Mice with Deficits in GFRalpha1 Signaling. J Neurosci Official J Soc Neurosci 2009, 29, 10695–10705, doi:10.1523/jneu- 781 rosci.2658-09.2009.

Author Response

1) What the horizontal bars in Fig 1C~E and Fig 4?

There are not identical between the horizontal bars and the calculated values in Table 1 and 2 

For example, 

 Table 2: Frequency sIPSCs-GDNF 3.6±0.4

 Fig 4A: horizontal bar for sIPSCs-gdnf was over "4"

The horizontal lines in the scatter plots denote the median of the cell-averaged measurements, while in the tables sthe data is presented as Mean ± Standard Error of Mean. We added the median values to the table, in parentheses after the Mean ± SEM, for clarity.

2) Figure 8

Authors conducted these experiments using tissue from 3 patients, who were quite different for age and sex, in Table 3.

Descriptions for "n=4 for controls and n=5 for GDNF" in footnote of Fig 8 confused many readers how select tissues from 3 patients. 

Furthermore, it sounds the individual difference between patients might be negligible for these experiments.

We apologize for not being clear on the definition of n for these experiments. The n=4 and n=5 values refer to number of cells recorded, not to the number of patients. We added the word “cells” after the number to further clarify this.

3) Reference styles have to recheck very carefully (Abbreviations, sometimes found "Is" etc)

Foe example " J Neurosci Official J Soc Neurosci" in ref 31 

  1. Canty, A.J.; Dietze, J.; Harvey, M.; Enomoto, H.; Milbrandt, J.; Ibáñez, C.F. Regionalized Loss of Parvalbumin Interneurons in the 780 Cerebral Cortex of Mice with Deficits in GFRalpha1 Signaling. J Neurosci Official J Soc Neurosci 2009, 29, 10695–10705, doi:10.1523/jneu- 781 rosci.2658-09.2009.

We thank the reviewer for spotting the mistake, there was an unexpected issue with some of the journal abbreviations in our reference manager system. We now re-checked the reference style for the whole manuscript and re-formatted them using the IJMS template.

Reviewer 2 Report (New Reviewer)

The study by Mikroulis et al. describe the mechanisms of neurotrophic factor GDNF participation in the inhibitory synaptic drive on principal neurons in the hippocampus of mice and human brain. The experiments seem to be well performed and the conclusions are based on the results. The manuscript is within the scope of International Journal of Molecular Sciences journal. I have, however, several comments that should be addressed to try to improve the presentation of the data:

1. In Table 1, please clarify whether “n“ is biological or technical repetitions?

2. Table 2. Why is DMSO listed as a solvent?

3. A number of previous published studies on the GDNF action connect its main physiological functions with RET phosphorylation and activation of subsequent intracellular mechanisms. Please indicate more clearly in the “Discussion” or “Introduction” section what is the novelty of the current study?

4. Please explain, why does Figure 5A represent a significant difference in RET phosphorylation under different conditions whereas Figure 5B shows no difference?

5. There is a necessity to specify the age of experimental animals when examining a GABAergic system. It is well known that experimental animals as well as humans do not have GABAergic system reversal at birth, and newborn animals may have different characteristics of inhibitory and excitatory system activity in the hippocampus.

6. Based on what data do the authors suggest that Syndecan3 expression level is related to GDNF?

7. In the Figure 8, “n” indicates the number of cells. What does “n” show in the other figures? Number of slices or animals? Please explain this statement in each figure legend. Authors are encouraged to think about the uniform presentation of data.

8. Why the authors did not perform inhibitory analysis when working with human slices?

Author Response

The study by Mikroulis et al. describe the mechanisms of neurotrophic factor GDNF participation in the inhibitory synaptic drive on principal neurons in the hippocampus of mice and human brain. The experiments seem to be well performed and the conclusions are based on the results. The manuscript is within the scope of International Journal of Molecular Sciences journal. I have, however, several comments that should be addressed to try to improve the presentation of the data:

  1. In Table 1, please clarify whether “n“ is biological or technical repetitions?

The number of cells recorded is presented as n in figures and tables throughout the manuscript. We added an explanation to the tables 1 and 2, and added an explanation in the method section:

“The number of cells is reported as n for all electrophysiology measurements, with one cell used per slice”

  1. Table 2. Why is DMSO listed as a solvent?

For the experiment presented in Table 2, we used XIB4035, which, according to the manufacturer (Sigma-Aldrich) must be dissolved in DMSO. So we dissolved the XIB4035 powder in DMSO to a 10 mM stock solution.

  1. A number of previous published studies on the GDNF action connect its main physiological functions with RET phosphorylation and activation of subsequent intracellular mechanisms. Please indicate more clearly in the “Discussion” or “Introduction” section what is the novelty of the current study?

The novelty of the study is in enhancement of inhibitory drive onto the pyramidal neurons by GDNF, which has not been known prior to this study, and can be important for better understanding the seizure-suppressant effects of GDNF observed in animal models. We mention this novelty aspect in several places in the discussion and introduction, but have strengthened some of the wording.

In the Introduction: “Although several hypotheses have been put forward, the current understanding of seizure-suppressant mechanisms of GDNF is rather limited. One possibility is that GDNF promotes survival of inhibitory interneurons, similar to what has been shown for dopaminergic neurons in the substantia nigra [18], or in molecular layer interneurons of the cerebellum [19]. Alternatively, GDNF might promote inhibition indirectly by other mechanisms, such as stimulating neurite outgrowth [20], or inhibiting microglia activation [21]. GDNF may exert an effect through Ret [2], NCAM [3] or Syndecan-3 [5] pathways, but which of these is involved in its seizure-suppressant effect are currently unknown”

In the Discussion: “Previous studies from our and other groups have shown an anti-seizure effect of GDNF when either delivered or over-expressed in epileptic tissue in animal models of epilepsy [15–17]. However, there was a gap in the understanding of the mechanisms of GDNF action. With this study, we addressed which of the different GDNF signaling mechanisms might contribute to its seizure-suppressant effects”.

And later: “In conclusion, here we identified a previously unknown mechanism of GDNF action enhancing inhibitory drive onto the hippocampal principal neurons. This novel mechanism can explain, at least partially, the seizure-suppressant effects of GDNF observed earlier in animal models”.

  1. Please explain, why does Figure 5A represent a significant difference in RET phosphorylation under different conditions whereas Figure 5B shows no difference?

The difference in RET phosphorylation was moderate (~20%) but significant when analyzed and normalized to internal controls. This might not be entirely clear from the images of the blots but was a consistent result through several rounds of experiments. Moderate differences such as this one might be difficult to judge by looking at intensity of bands in the blots alone by eye, as even slight changes in intensity of normalization bands for example would result in significant relative differences.

  1. There is a necessity to specify the age of experimental animals when examining a GABAergic system. It is well known that experimental animals as well as humans do not have GABAergic system reversal at birth, and newborn animals may have different characteristics of inhibitory and excitatory system activity in the hippocampus.

All animals used were no less than 3 weeks of age and no more than 10 weeks of age. Please see Methods 4.3 Slice preparation. At this age, GABA released from interneurons in the hippocampus is already inhibitory and show the same characteristics as in older mice (Murata et al., Science Advances 2020, DOI: 10.1126/sciadv.aba1430).

  1. Based on what data do the authors suggest that Syndecan3 expression level is related to GDNF?

The experiments on Syndecan3 expression were performed to clarify the location of its expression in the CA1 area especially with regards to synapses, not to correlate Syndecan3 expression levels with GDNF exposure. If Syndecan3 was found to be expressed at synapses in the pyramidal cell layer of CA1, where perisomatic inhibitory synapses are present, then the contribution of Syndecan3 signaling on the effects observed in IPSCs could not have been excluded. However, we find that Syndecan3 is mainly expressed in what appears to be axons in the stratum radiatum, with very low colocalization with synaptic terminal markers, and therefore its contribution to the effects of GDNF we observe is less likely.

  1. In the Figure 8, “n” indicates the number of cells. What does “n” show in the other figures? Number of slices or animals? Please explain this statement in each figure legend. Authors are encouraged to think about the uniform presentation of data.

For all electrophysiology measurements in the figures and tables, the number of cells is indicated as n, from an equal number of slices, as only one cell was recorded from each slice. We added a clarification in Methods 4.9 Quantification and statistical analysis.

“The number of cells is reported as n for all electrophysiology measurements, with one cell used per slice”

  1. Why the authors did not perform inhibitory analysis when working with human slices?

The analysis on human slices was done on inhibitory currents, similarly to what performed in mouse slices. The figure legend and the corresponding paragraph already state that analysis was made on “IPSCs”.

Round 2

Reviewer 1 Report (New Reviewer)

The revised manuscript is acceptable to publish on the journal.

This manuscript is a resubmission of an earlier submission. The following is a list of the peer review reports and author responses from that submission.

Round 1

Reviewer 1 Report

The Authors of the manuscript entitled “GDNF increases inhibitory synaptic drive on principal neurons in the hippocampus via activation of the Ret pathway” aim to describe the mechanisms by which GDNF counteracts seizures when its concentration is increased in the hippocampus. The Authors address this question by analyzing inhibitory and excitatory sIPSCs and mIPSCs in mouse and human hippocampus, along with immunostaining analysis. The Authors state that GDNF increases the inhibitory signaling impinging principal hippocampal neurons, by a RET-mediated pathway “likely” acting postsynaptically.

The problem addressed by the study is highly relevant, in view of a future possibility to translate GDNF to human therapy. However, the manuscript has several major flaws precluding publication in the present form.

Major points

  1. All the manuscript is written in an unusual dubitative form. Too many “seem” verbs in the text, along with several other dubitative terms (would, could, likely, non-significant trend, etc.), not only in the Discussion. Indeed, the data presented do not allow to make stronger statements, so probably the dubitative form is justified by data, but then the paper is not justified by data. In any case, all the hypotheses should be discussed in the Discussion, and not highlighted in the Results section.
  2. Human data are very weak. No traces are shown, few cells analyzed, and, strikingly, bad conclusion derived from the shown data. In particular, the Authors state the GDNF, administered in human hippocampal slices, induces a “slight increase” of the sIPSC amplitude, while in fig. 8 it is extremely clear that in the presence of GDNF the sIPSC amplitude decreases. This finding is in strong contrast with the mouse data and weakens the meaning of the whole paper.
  3. GDNF, postsynaptic vs presynaptic mechanisms: the Authors state that the data suggest a presynaptic mechanism, but I think this conclusion is too strong. It is true that an amplitude increase is usually linked to a presynaptic phenomenon, but here we see that GDNF affects clearly more frequency than amplitudes. When you observe a mIPSC frequency increase, you cannot exclude the involvement of a presynaptic mechanism.
  4. The pharmacological analysis of the involvement of RET in the GDNF effects on inhibitory signalling is weak. Both XIB4035 and SPP86 display contrasting effects in different experiments. Some action of SPP86 is declared by the Authors as non-specific (SPP86 increases the IPSC frequency), and data are not further discussed. Probably it would be better to eliminate the SPP86 data from the paper.
  5. The Authors’ statement about the peri-somatic effects of GDNF is mainly based on current kinetics, which could change for other reasons than distance from the electrode. Images showing gephyrin puncta are not so nice and the reason to remove perinuclear puncta is not sufficiently clarified.

Minor points

  1. In the Introduction, from line 69 to 73, the argument about a “certain threshold” is not clear and should be rephrased.
  2. In the Discussion, the argument on GDNF actions (line 396-408) is not properly linked with the surrounding text.
  3. In tables, please use Hz for frequency.
  4. In all tables, please use fewer digits after the comma: e.g., 30.260±0.405 should be 30.3±0.4.
  5. It would be nice to use the same axes scaling for graphs illustrating frequency, amplitudes and rise times for sIPSCs and mIPSCs.
  6. In fig. 6, please label the figure panels, to facilitate legibility.
  7. In figure 7B, please create an ordinate label.
  8. In Methods, please substitute cerebellum, with hippocampus (Line 444).
  9. Why do the Authors wait 24 hours before using the human slices? (Line 466)
  10. Why and where do the Authors use PTX? (Line 489).
  11. In general, it would be better to have real P values instead of p<…, when possible.

Author Response

Reviewer 1 comments:

The Authors of the manuscript entitled “GDNF increases inhibitory synaptic drive on principal neurons in the hippocampus via activation of the Ret pathway” aim to describe the mechanisms by which GDNF counteracts seizures when its concentration is increased in the hippocampus. The Authors address this question by analyzing inhibitory and excitatory sIPSCs and mIPSCs in mouse and human hippocampus, along with immunostaining analysis. The Authors state that GDNF increases the inhibitory signaling impinging principal hippocampal neurons, by a RET-mediated pathway “likely” acting postsynaptically.

The problem addressed by the study is highly relevant, in view of a future possibility to translate GDNF to human therapy. However, the manuscript has several major flaws precluding publication in the present form.

Major points

  1. All the manuscript is written in an unusual dubitative form. Too many “seem” verbs in the text, along with several other dubitative terms (would, could, likely, non-significant trend, etc.), not only in the Discussion. Indeed, the data presented do not allow to make stronger statements, so probably the dubitative form is justified by data, but then the paper is not justified by data. In any case, all the hypotheses should be discussed in the Discussion, and not highlighted in the Results section.

Response: We reviewed the language in the whole manuscript and removed some dubitative forms especially in the Result and Discussion sections.

  1. Human data are very weak. No traces are shown, few cells analyzed, and, strikingly, bad conclusion derived from the shown data. In particular, the Authors state the GDNF, administered in human hippocampal slices, induces a “slight increase” of the sIPSC amplitude, while in fig. 8 it is extremely clear that in the presence of GDNF the sIPSC amplitude decreases. This finding is in strong contrast with the mouse data and weakens the meaning of the whole paper.

Response: We have now added representative traces for recordings from human slices in the new Figure 8. We thank the reviewer for pointing out our mistake, it is true that GDNF in human slices induces a slight decrease in amplitude of sIPSCs, as clearly shown in the figure. This has now been corrected in the revised manuscript.

It is true that the finding in IPSC amplitude changes is in contrast with the mouse data. However the shift in the distribution of inter event intervals was consistent with mouse data, and showed higher frequency of sIPSCs after GDNF incubation. The reason behind this difference is currently unknown, but we have previously observed differences in effects of neuropeptides between rodent and human tissue (see Ledri et al., Journal of Neuroscience, 2015). While it is possible that GDNF mechanisms might differ or be more complex in the human slices, it is also worth noting that the general quality and health of slices from resected tissue is variable compared to acute mouse slices. This might also contribute to the difficulties in some of the results, especially when only few surgical resections per year are available.

Nevertheless, we believe that at least partial consistency with the mouse data is of interest and warrants the inclusion of human data in the manuscript. Further research is required to identify in more details the mechanisms of GDNF action in human epileptic tissue.

We have now added a paragraph in the discussion (from line 360) to clarify these points:

“Increased inhibitory synaptic drive was also partially maintained in human hip-pocampal slices. While results of IPSC amplitude changes seem to be is in contrast with the mouse data, the shift in the distribution of inter event intervals was consistent and showed higher frequency of sIPSCs after GDNF incubation. The reason behind this difference is currently unknown, but we have previously observed differences in effects of neuro-peptides between rodent and human tissue [35]. While it is possible that GDNF mech-anisms might differ or be more complex in the human slices, it is also worth noting that the general quality and health of slices from resected tissue is variable compared to acute mouse slices. This might contribute to the differences observed here”.

  1. GDNF, postsynaptic vs presynaptic mechanisms: the Authors state that the data suggest a presynaptic mechanism, but I think this conclusion is too strong. It is true that an amplitude increase is usually linked to a presynaptic phenomenon, but here we see that GDNF affects clearly more frequency than amplitudes. When you observe a mIPSC frequency increase, you cannot exclude the involvement of a presynaptic mechanism.

Response: We believe the reviewer here means “postsynaptic mechanism”, as that is what we suggest in our manuscript. It is true that GDNF affects both frequency and amplitude of IPSCs, and while this might also suggest pre-synaptic mechanisms the concomitant increase in gephyrin immunoreactivity supports the notion that post-synaptic mechanisms are more likely explanation. We however agree with the reviewer that with an increase in frequency of mIPSC we cannot completely exclude pre-synaptic mechanisms, and have therefore added a sentence in the revised manuscript to include this possibility:

“Such simultaneous changes in both frequency and amplitude of synaptic events is commonly interpreted as alterations at the postsynaptic site [23], although pre-synaptic mechanisms cannot completely be excluded.”

  1. The pharmacological analysis of the involvement of RET in the GDNF effects on inhibitory signalling is weak. Both XIB4035 and SPP86 display contrasting effects in different experiments. Some action of SPP86 is declared by the Authors as non-specific (SPP86 increases the IPSC frequency), and data are not further discussed. Probably it would be better to eliminate the SPP86 data from the paper.

Response: We agree with the reviewer that the non-specific nature of SPP86 might complicate interpretation of some results. Especially when investigating synaptic changes with electrophysiology, which is sensitive to detect small alterations in complex pathways. We therefore removed the SPP86 electrophysiological data, but kept the Western Blots as we believe that in those conditions, we can convincingly show that, although in a non-specific manner, SPP86 is still able to prevent Ret phosphorylation by GDNF. This result at least strengthens the idea that in our slices GDNF is able to phosphorylate Ret. Taken together with the inability of GDNF to phosphorylate Fyn (downstream effect of NCAM activation) in the same assay, strongly suggests that Ret pathway is the main one involved in the described effects.

  1. The Authors’ statement about the peri-somatic effects of GDNF is mainly based on current kinetics, which could change for other reasons than distance from the electrode. Images showing gephyrin puncta are not so nice and the reason to remove perinuclear puncta is not sufficiently clarified.

Response: The kinetics of IPSC rise times are generally accepted to be reflecting the distance from the electrode and can therefore be used to discern perisomatic or dendritic events. While it is true that GABA-A mediated current kinetics can change for other reasons, as for example differences in composition of GABA-A receptor subunits, these typically alter current decay (Bosman et al., Journal of Neurophysiology 2005; Ramadan et al., Journal of Neurophysiology 2003). In addition, significant alterations in GABA-A receptor composition are unlikely to happen substantially within the 1 hour time window of GDNF incubation, as they would require signaling, gene transcription, protein expression, assembly and membrane translocation.

Regarding the removal of perinuclear, intracellular gephyrin puncta from the analysis, it has been previously shown how gephyrin can accumulate in intracellular locations when already anchored to receptors on their way to the plasma membrane (Hanus et al., Jounal of Neuroscience 2004, also ref. 20 in the manuscript). This is most likely what we also observe in our images. Nevertheless, since the aim was to evaluate whether increased amplitude and frequency of IPSC could be explained by an increase in active post-synaptic GABA-A receptors, we focused our analysis only on gephyrin puncta in the proximity of the plasma membrane. While we did not have a separate labeling of the membrane, we therefore decided to exclude only puncta overlaid with the cell nucleus, which most likely do not represent synapses.

Minor points

  1. In the Introduction, from line 69 to 73, the argument about a “certain threshold” is not clear and should be rephrased.

Response: To clarify, we removed ”certain” from the sentence.

  1. In the Discussion, the argument on GDNF actions (line 396-408) is not properly linked with the surrounding text.

Response: We thank the reviewer for the suggestion. The paragraph has now been moved to better link with the surrounding text (lines 391-405 in the revised manuscript).

  1. In tables, please use Hz for frequency.

Response: We changed 1/s to Hz in all tables.

  1. In all tables, please use fewer digits after the comma: e.g., 30.260±0.405 should be 30.3±0.4.

Response: We now decreased number of digits after the comma in all tables.

  1. It would be nice to use the same axes scaling for graphs illustrating frequency, amplitudes and rise times for sIPSCs and mIPSCs.
  2. In fig. 6, please label the figure panels, to facilitate legibility.
  3. In figure 7B, please create an ordinate label.

Response: These figures have now been corrected according to the reviewer’s suggestions.

  1. In Methods, please substitute cerebellum, with hippocampus (Line 444).

Response: We apologize for the confusion. We changed the term ”excised” with “removed”. We remove the cerebellum before proceeding with slicing the hemispheres.

  1. Why do the Authors wait 24 hours before using the human slices? (Line 466)

Response: We typically wait 24 hours before recording from human slices because we have recently developed a method to maintain them for up to 48 hours with minimal impact on cell survival and quality of electrophysiological characteristics of the cells and slices (see Wickham et al., Scientific Reports 2018, added now to the manuscript). This greatly enhances and improves the work with such tissue, which was previously only viable for 12-16 hours.

  1. Why and where do the Authors use PTX? (Line 489).

Response: We used PTX for isolating and recording the excitatory PSCs presented in Supplementary Figure 1.

  1. In general, it would be better to have real P values instead of p<…, when possible.

Response: This has now been corrected in the revised manuscript for some analyses. Real P values are reported for example in the tables. However, in most of our K-S comparisons, P values were really low, some being in the range of 1E-80 or lower. We believe that reporting them as such would be unusual and possibly be also confusing to some readers.

Reviewer 2 Report

  1. The paper structure should be described at the end of the introduction section.
  2. The introduction is too weak. The introduction should indicate the research gaps and research goals.
  3. The authors are suggested to use a flowchart to describe the proceeding flowchart in the Methodology section. Please use a standard flowchart to illustrate the process. For example, use an elliptical graph to illustrate the “start” and “end”, use the diamond graph to illustrate the judgment events.
  4. How about change the sequence of section 2 result and section 3 methodology section?
  5. The authors are suggested to list that you have solved the research gaps. What is the main question addressed by the research? Your contributions should be presented in conclusion.
  6. No comparison with related works, it is hard to persuade readers. What does it add to the subject area compared with other published material?
  7. Are the conclusions consistent with the evidence and arguments presented and do they address the main question posed?
  8. Please include any additional comments on the tables and figures.

Author Response

Reviewer 2 comments:

  1. The paper structure should be described at the end of the introduction section.

Response: We now modified the last sentence of the introduction to provide additional details on the manuscript structure and content:

“Here, by using a combination of electrophysiology, western blot and imaging techniques, we first demonstrate that elevated extracellular levels of GDNF increase inhibitory synaptic drive on principal neurons in mouse and human acute hippocampal slices. We then provide evidence that these effects are most likely postsynaptic, and that GDNF acts preferentially via Ret-dependent pathway”.

  1. The introduction is too weak. The introduction should indicate the research gaps and research goals.

Response: We have now expanded the introduction to better frame our research questions in the context of the current knowledge gaps in this area:

“Although several hypotheses have been put forward to explain GDNF’s action, current understanding of seizure-suppressant mechanisms of GDNF is rather limited. One possibility is that GDNF promotes survival of inhibitory interneurons, similarly to how it can protect dopaminergic neurons in the substantia nigra [18], or in molecular layer inter-neurons of the cerebellum [19]. Alternatively, GDNF might promote inhibition indirectly by other mechanisms, such as stimulating neurite outgrowth [20], or inhibiting microglia activation [21]. GDNF may exert an effect through Ret [2], NCAM [3] or Syndecan-3 [5] pathways, but which of these are involved in its seizure-suppressant effect is currently unknown”.

  1. The authors are suggested to use a flowchart to describe the proceeding flowchart in the Methodology section. Please use a standard flowchart to illustrate the process. For example, use an elliptical graph to illustrate the “start” and “end”, use the diamond graph to illustrate the judgment events.

Response: We have now added a flowchart in the method section (Figure 10).

  1. How about change the sequence of section 2 result and section 3 methodology section?

Response: Journal formatting requires the Results and Methodology sections to be in the order presented in the manuscript.

  1. The authors are suggested to list that you have solved the research gaps. What is the main question addressed by the research? Your contributions should be presented in conclusion.

Response: We now added a concluding sentence which refers to the research gaps we identified in the introduction section:

“In conclusion, here we identified a previously undefined mechanisms of action by which GDNF promotes inhibitory drive in the hippocampus through activation of the Ret pathway. These mechanisms of action might contribute to the seizure-suppressant effects of GDNF observed earlier in animal models, stimulate further research in the field and ultimately promote the development of GDNF-based therapies against epilepsy. Based on these and previous results, one might therefore envisage a clinical scenario whereby overexpression of GDNF in the hippocampus enhances inhibition of principal neurons and thereby counteracts seizure-like activity”.

  1. No comparison with related works, it is hard to persuade readers. What does it add to the subject area compared with other published material?
  2. Are the conclusions consistent with the evidence and arguments presented and do they address the main question posed?

Response: Please see our response to the previous comments (points 1, 2 and 5). We have now modified both the introduction and discussion sections to better frame our work in the context of previous research.

  1. Please include any additional comments on the tables and figures.

Response: We do not have additional comments or modifications on Figures and Tables. However, some figures were adjusted according to comments from the editor and reviewer 1.

Round 2

Reviewer 1 Report

The revised version of the paper is slightly improved but not substantially improved. The manuscript is still very weak. Many results are still presented in a dubitative form, and conclusions are weak. Some non-exhaustive examples of dubitative sentences:

Should 72

Most likely 105

It seemed like 118

This observation fits well 133

This could 173

Would 197

May reflect 202

Non-significant trend 227

Most likely 284

Can be assumed 330

Could also be 334

Support the idea 344

Despite the Authors’ reply to this specific point, it is very difficult to assess that the described phenomena “is most likely postsynaptic”, when “presynaptic mechanisms cannot be excluded”. The increase of IPSCs frequency is clearly a presynaptic mechanism, and the Authors show amplitude increase in mice and decrease in human.

Despite Authors’ reply, the electrophysiological data suggesting, based on kinetics, peri-somatic localization of strengthened synapses are weak.

Reviewer 2 Report

The authors have fixed previous concerns.